# CADMorph: Geometry-Driven Parametric CAD Editing via a Plan-Generate-Verify Loop

**Weijian Ma**[*]
Fudan University
weijian.ma1@gmail.com

**Shizhao Sun**[†]
Microsoft Research, Asia
shizsu@microsoft.com

**Ruiyu Wang**
University of Toronto
rwang@cs.toronto.edu

**Jiang Bian**
Microsoft Research, Asia
jiabia@microsoft.com

## Abstract

A Computer-Aided Design (CAD) model encodes an object in two coupled forms: a *parametric construction sequence* and its resulting *visible geometric shape*. During iterative design, adjustments to the geometric shape inevitably require synchronized edits to the underlying parametric sequence, called *geometry-driven parametric CAD editing*. The task calls for 1) preserving the original sequence's structure, 2) ensuring each edit's semantic validity, and 3) maintaining high shape fidelity to the target shape, all under scarce editing data triplets. We present *CADMorph*, an iterative *plan–generate–verify* framework that orchestrates pretrained domain-specific foundation models during inference: a *parameter-to-shape* (P2S) latent diffusion model and a *masked-parameter-prediction* (MPP) model. In the planning stage, cross-attention maps from the P2S model pinpoint the segments that need modification and offer editing masks. The MPP model then infills these masks with semantically valid edits in the generation stage. During verification, the P2S model embeds each candidate sequence in shape-latent space, measures its distance to the target shape, and selects the closest one. The three stages leverage the inherent geometric consciousness and design knowledge in pretrained priors, and thus tackle structure preservation, semantic validity, and shape fidelity respectively. Besides, both P2S and MPP models are trained without triplet data, bypassing the data-scarcity bottleneck. CADMorph surpasses GPT-4o and specialized CAD baselines, and supports downstream applications such as iterative editing and reverse-engineering enhancement.

## 1 Introduction

Computer-Aided Design (CAD) serves as the vital bridge between an initial concept and a manufacturable product. A CAD model thus exhibits a **representational duality**, carrying two tightly coupled representations of the same object. The *parametric construction sequence* (left in Figure 1(a)) defines exact spatial relationships through operations (e.g., Line and Extrude) and numeric parameters (e.g., 223 and 128), ensuring manufacturing accuracy while preserving full editability for future revision. The *visible geometric shape* (right of Figure 1(a)) is rendered from that sequence, providing an intuitive and universally understood visual reference for inspection, simulation and validation.

During CAD development, the geometric shape is frequently adjusted—whether to satisfy simulation feedback, ergonomic requirements or aesthetic goals—which in turn demands corresponding edits to

---

[*]Work done during internship at Microsoft Research, Asia.
[†]Corresponding author.

39th Conference on Neural Information Processing Systems (NeurIPS 2025).

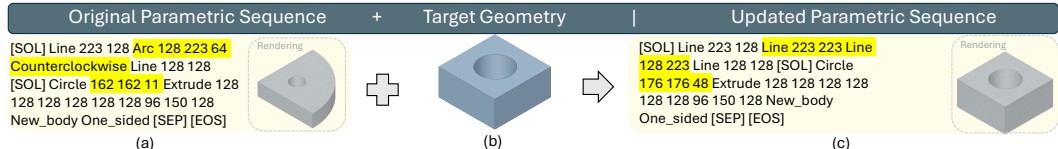

Figure 1: Given a **parametric construction sequence** (left) and a **target geometry-only shape** (centre), CADMorph outputs an **updated construction sequence** (right) whose rendering matches the target while maintaining the pattern of the original construction sequence. Renderings in grey are shown only for visual reference. Edits in the sequences are highlighted. An intentionally simple example is shown here for clarity. More complex results appear in Section 4.2 and the Appendix.

the underlying parametric sequence, the authoritative source for manufacturing. We call this process **geometry-driven parametric CAD editing**: given an *original parametric sequence* and a *target geometric shape*, generate an *updated parametric sequence* that reproduces the target geometric shape (Figure 1). This task is laborious and error-prone, as engineers must assess the magnitude of shape changes, pinpoint the exact segments to modify within a complex and nested parametric sequence, and then propagate those changes through all dependent segments.

Despite its practical importance, geometry-driven parametric editing remains largely unexplored. Most prior work addresses *unconditional editing* [Xu et al., 2022, 2023, Zhang et al., 2024], which generates an edited parametric sequence from an original parametric sequence without any explicit guidance. A few studies explore *text-based editing* [Yuan et al., 2025], which uses a textual instruction alongside the original parametric sequence to drive edits. However, expressing comprehensive shape changes through a single textual instruction is often cumbersome and unintuitive for end users.

The task of geometry-driven parametric CAD editing presents several core challenges. The first, *structure preservation*, demands that edits be confined to those segments of the original parametric sequence responsible for the desired shape change, leaving all other parts untouched. The second, *semantic validity*, requires that the updated parametric sequence not only be syntactically correct but also yield a realistic CAD model that adheres to design conventions——such as evenly distributing bolt holes rather than placing them arbitrarily. The third, *shape fidelity*, mandates that the updated parametric sequence, when rendered, reproduce the target shape. In addition, *data scarcity* poses a fundamental obstacle, since no existing dataset combines an original parametric sequence with both a target geometric shape and its corresponding updated sequence.

To tackle these challenges, we leverage the geometric and design priors insde the domain-specific pre-trained foundation models, and introduce **CADMorph** (Figure 2), an iterative *plan–generate–verify* framework that incrementally transforms the original parametric sequence into one that reproduces the target geometry by employing two complementary models, the *parameter-to-shape model (P2S)* and the *masked-parametric-prediction (MPP) model*. The P2S is a latent diffusion model (LDM) [Rombach et al., 2022] trained to map a parametric sequence into its corresponding geometric shape, while the MPP model is a Large Language Model [Touvron et al., 2023, Meta AI, 2024] trained to infill masked segments of the parametric sequence. Neither model relies on scarce triplet data—⟨original sequence, target geometry, updated sequence⟩——thereby sidestepping the *data-scarcity* bottleneck.

The editing is an iterative synchronization between P2S and MPP. In each iteration, the *planning* stage identifies which segments in the current parametric sequence to modify. Inspired by advances in text-to-image LDMs [Rombach et al., 2022, Hertz et al., 2022a], we analyze cross-attention maps inside P2S to quantify segment-wise influence on the target geometric shape. The segments no longer contributes to target geometric shape are deemed safe to edit and replaced by a special [mask] token. This masking strategy confines edits into useless segments, thereby preserving the useful segments of the sequence and satisfying the *structure-preservation* requirement. The *generation* stage proposes candidate edits by changing identified segments only. We utilize the MPP model to infill each [mask] with suitable edits. As the model is trained on large-scale parametric sequence data, it produces syntactically correct and semantically meaningful edits, ensuring *semantic validity*. The *validation* stage selects the candidate parametric sequence that best matches the target geometric shape. We map both the candidate parametric sequences and the target geometric shape into a shared space, i.e., the latent space of the P2S model. Distances in this shared space serve as an efficient proxy for geometric

dissimilarity. The candidate of minimal distance to the target geometric shape is retained for the next iteration, driving convergence toward the target geometric shape and thereby enforcing *shape fidelity*.

CADMorph offers several key advantages. First, it is data-efficient. Rather than relying on labor-intensive triplet supervision, it allocates effort to inference time by running the P2S and MPP models in its plan–generate–verify cycle. This strategy mirrors the principle of test-time scaling [Cobbe et al., 2021, Ma et al., 2025], where additional computation during inference yields significantly improved results. Second, it is exploration-efficient. By leveraging P2S model in the planning stage, it reduce the search space of possible edits; by using P2S model again in the verification stage, it provides an effective signal that steers the edit toward a promising direction.

Experiments demonstrate that CADMorph outperforms state-of-the-art general-purpose models (GPT-4o [OpenAI, 2024]) and powerful CAD-specific baselines [Ma et al., 2024, Zhang et al., 2024] both quantitatively and qualitatively. In addition, we showcase two downstream applications—iterative geometry editing and reverse-engineering refinement—highlighting CADMorph's versatility in real-world design workflows. Our key contributions are summarized as follows:

- We formalize geometry-driven parametric CAD editing, a practical task in CAD workflow. It takes an original parametric sequence and a target geometric shape as input and produces an updated parametric sequence whose rendered shape matches the target.
- We leverage two complementary models, a parameter-to-shape (P2S) diffusion model and a masked-parametric prediction (MPP) Transformer, sidestepping the data-scarcity bottleneck.
- We introduce an iterative plan-generate-verify framework that jointly exploits P2S and MPP models. We analyze P2S model's cross-attention maps to localize segments requiring edits (planning), use MPP model to infill those segments with semantically valid candidate revisions (generation), and embed each candidate and the target shape in the P2S model's latent space to select the sequence whose rendering closely matches the target shape (verification).

## 2 Related Work

**Deep Learning in CAD.** Since the pioneering work of Wu et al. [2021], deep learning research in Computer-Aided Design (CAD) has witnessed a surge and now spans four main directions: representation learning, generation, reverse engineering, and editing. Representation learning methods derive compact embeddings, either unimodal [Wu et al., 2021] or multimodal [Ma et al., 2023], to support downstream recognition tasks. Generation approaches translate free-form text into CAD parametric sequences [Khan et al., 2024b, Wang et al., 2025c, Li et al., 2024]). Reverse-engineering pipelines reconstruct editable primitives [Uy et al., 2022, Li et al., 2023, Ren et al., 2022] or full parametric sequences [Khan et al., 2024a, Ma et al., 2024, Lambourne et al., 2022] from raw 3D geometry. Editing methods begin with an original parametric sequence and output a revised one, either via random sampling [Xu et al., 2022, 2023, Zhang et al., 2024] or under text-driven guidance [Yuan et al., 2025]. Our focus, geometry-driven parametric editing, shares surface similarities with previous reverse engineering and editing methods, yet differs in essential ways. Reverse-engineering pipelines ignore the designer's intent embodied in the original parametric sequence, while prior editing methods disregard the rich visual cues provided by a target geometry shape. By simultaneously respecting the original sequence and leveraging the target shape as guidance, our work bridges this gap and addresses a problem that neither existing reverse-engineering nor editing methods fully resolve.

**3D Shape Editing.** Research in this area operates directly on the geometry shape, whether as mesh, signed distance field (SDF) or neural radiance field (NeRF). These methods offer part-aware, NeRF-based, mesh-guided, coupled-optimization, and SDF-sculpting frameworks for 3D shape editing [Hertz et al., 2022b, Wang et al., 2023b, Chen et al., 2023, Wang et al., 2023a, Hu et al., 2024, Rubab and Tong, 2025]. Besides, pretrained text-to-image diffusion models have been used to guide 3D shape editing, e.g., integrating 2D inpainting or score-distillation sampling (SDS) losses to achieve multi-view-consistent modifications [Yu et al., 2023, Jiang et al., 2023, Zhou et al., 2023, Dihlmann et al., 2024, Poole et al., 2022]. In contrast, our work treats the shape as guidance and performs edits on the underlying parametric sequence, rather than directly deforming the shape itself.

**Test-time Scaling with Verifiers.** The core idea is to increase inference-time effort, i.e., generating multiple candidate outputs and then using a verifier to select the best one, a paradigm shown to substantially improve generative models [Cobbe et al., 2021]. This strategy has been applied across domains, e.g., large language models, vision-language models, image generation, speech synthesis and video generation [Lee et al., 2025, Wang et al., 2025b, Dong et al., 2023, Wang et al., 2025a,

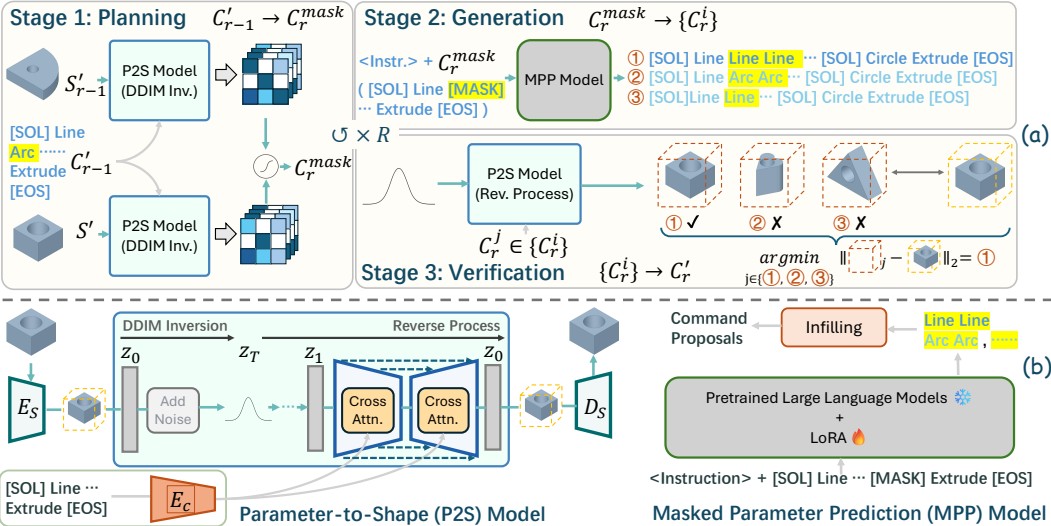

Figure 2: General pipeline of CADMorph. **(a)** Iterative editing loop. For each round $r \in R$: (i) The planning stage selects the editing location via peeking the cross-attention map of the P2S model. (ii) The generation stage propose candidate sequences via the MPP model (a finetuned LLM). (iii) The verification stage selects the candidate sequence best matches the target shape for round $r$. The parameters of each primitive in $C$ are omitted for brevity. **(b)** Architecture of P2S and MPP model.

Xie et al., 2025, Zhang et al., 2025, Chen et al., 2025, Li et al., 2025a, Ye et al., 2025, Peng et al., 2025, Cong et al., 2025, Li et al., 2025b]. In our approach, the generation stage invokes the MPP model multiple times to generate diverse candidate parametric sequences, and the verification stages examine their embeddings in latent space of the P2S model to select the sequence that best matches the target geometry shape. This procedure mirrors the principle of test-time scaling with verifiers, and is the first application of this paradigm to CAD related tasks.

# 3 Method

We first formalize the task of geometry-driven parametric CAD editing (Section 3.1). Then, we present an overview for CADMorph, our iterative plan-generate-verify framework that leverages the interplay between two complementary pretrained foundation models in CAD domains: the parameter-to-shape (P2S) model and the masked-parameter-prediction (MPP) model (Section 3.2). Next, we introduce the architecture of the P2S and MPP model (Section 3.3). Finally, we delve into each stage of the plan-generate-verify pipeline (Section 3.4).

## 3.1 Task Formulation

For a given CAD model, let $C$ denote its parametric construction sequence and $S$ its visual geometric shape. Given an original parametric sequence $C$ and a target geometric shape $S'$, *geometry-driven parametric CAD editing* seeks an updated parametric sequence $C'$ that reproduces $S'$ (Figure 1). Although many parametric sequences may reproduce $S'$, the preferred solution is the one that preserves the structure of $C$ rather than replacing it with an entirely different sequence. Thus, the overall goal is:

$$C' = \arg\min_{C'} \mathcal{D}_{\text{geometry}}(\mathcal{F}(C'), S') + \lambda \mathcal{R}_{\text{structure}}(C', C). \tag{1}$$

Here, $\mathcal{F}(C')$ denotes the rendered geometric shape of the sequence $C'$, either via CAD kernels or neural networks. $\mathcal{D}_{\text{geometry}}(\cdot, \cdot)$ measures the discrepancy between two geometric shapes, and $\mathcal{R}_{\text{structure}}(\cdot, \cdot)$ measures the structure similarity between two parametric sequences. $\lambda$ is an illustrative hyperparameter.

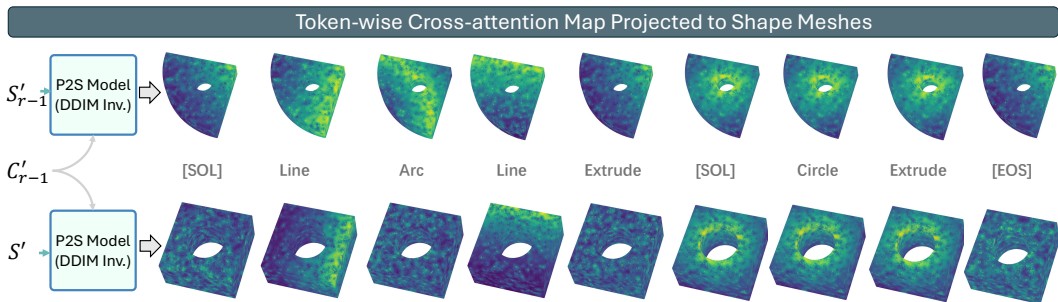

Figure 3: Visualization of P2S model's cross-attention (CA) maps. For each segment in the parametric sequence $C'_{r-1}$, we project its CA score on the derived mesh of its corresponding shape $S'_{r-1}$ (top row) and the target shape $S'$ (bottom row). Bright regions highlight the geometry that is strongly associated with the corresponding sequence segment, indicating that a part of the shape is more attracted to the sequence segment that describes it.

## 3.2 Approach Overview

CADMorph orchestrates two complementary pretrained foundation models (Figure 2(b)), a *parameter-to-shape (P2S)* model and a *masked-parameter-prediction (MPP)* model. The P2S model is a latent diffusion model trained on ⟨parametric sequence, shape rendering⟩ pairs, which translates a parametric sequence to a geometric shape. The MPP model is a Large Language Model (LLM) trained on massive construction sequences, which infills the masked region of plausible parametric sequences. Neither model requires scarce ⟨original sequence, target shape, updated sequence⟩ triplets for training.

Built on these models, CADMorph incrementally transforms the original parametric sequence $C$ into the sequence $C'$ that reproduces the target geometry $S'$ by iterating over three stages: *planning*, *generation* and *verification* (Figure 2(a)). At the $r$-th iteration:

1. Planning. Starting from the parametric sequence from the last iteration $C'_{r-1}$ ($C'_0 = C$), we locate the segments requiring edits by analyzing the P2S model's cross-attention maps. Those segments are replaced with the special token [mask], yielding the masked parametric sequence $C^{\text{mask}}_r$.

2. Generation. The MPP model infills the [mask] tokens $N$ times, producing a set of candidate sequences $C^1_r, \ldots, C^N_r$ that propose semantically plausible edits.

3. Verification. Each candidate is projected into the P2S model's latent space, and the one closest to the target geometric shape $S'$ is chosen as $C'_r$. This sequence seeds the next iteration.

The loop terminates when $C'_r$ converges or a maximum number of iterations is reached, yielding the final parametric sequence $C'$ that reproduces the target shape $S'$. For the details of each stage, please refer to Section 3.4

## 3.3 Model Architecture

CADMorph involves two foundation models pretrained on existing data in CAD domain. These models are originally designed for other tasks and are jointly used in our plan-generate-verify framework to achieve the geometry-driven parametric CAD editing task.

**Parameter-to-Shape (P2S) Model** (left of Figure 2(b)). We represent each shape as a voxelized truncated signed distance field (tSDF). The P2S model transforms parametric sequences into 3D shape latents, and further decodes it into SDF. It follows SDFusion's architecture [Cheng et al., 2023], and is re-trained on ⟨parametric sequences, SDF⟩ pairs of CAD data. It comprises two components: (1) a shape encoder-decoder pair ($E_s$ and $D_s$) that embeds SDFs into a latent space and reconstructs them, and (2) a diffusion model that maps parametric sequence into 3D shape latents.

**Masked-Parameter-Prediction (MPP) Model** (right of Figure 2(b)). It infills the masked segments of a parametric sequence. It adopts the the architecture of FlexCAD [Zhang et al., 2024], i.e., a LoRA-finetuned Llama-3 [Meta AI, 2024] 8B model with hierarchical masking strategies.

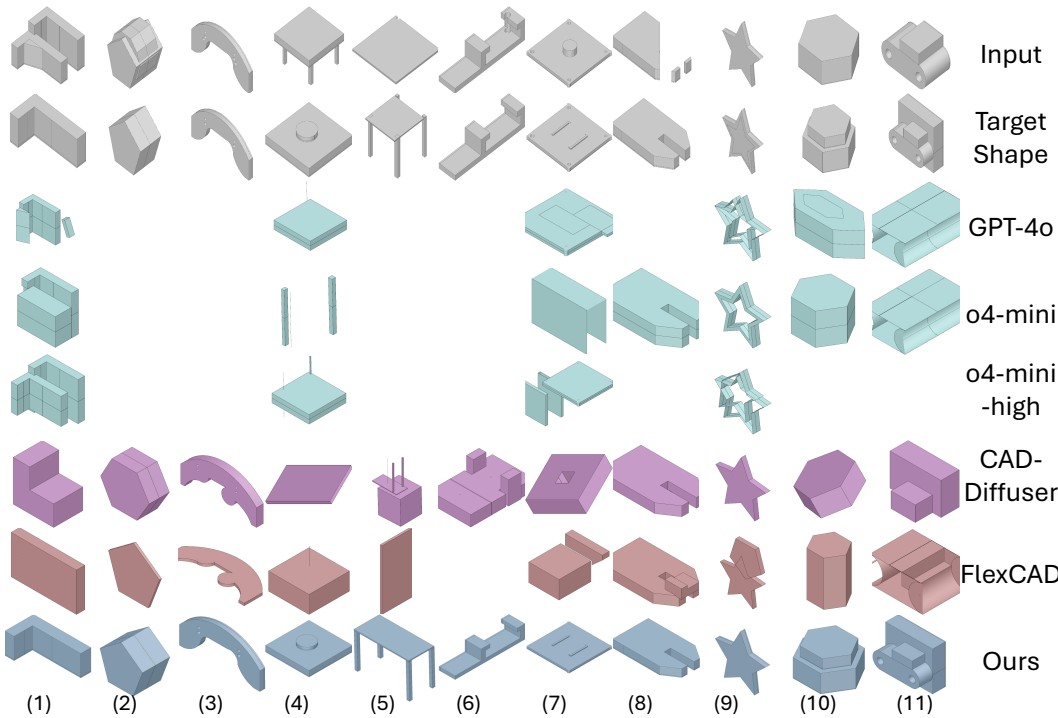

Figure 4: Qualitative results. **Top row:** rendering of original parametric sequence (shown instead of tokens for clarity). **Second row:** target shape. **Other rows:** renderings of the edited sequences from each method; empty cells indicate that no valid shape could be rendered. Better view zoomed in.

### 3.4 The Plan-Generate-Verify Framework

**Planning.** It receives the previous parametric sequence $C'_{r-1}$ and returns a masked parametric sequence $C^{\text{mask}}_r$, where each special [mask] token marks a segment requiring edits. A naive strategy would mask segments at random, but this wastes computation on portions of $C'_{r-1}$ that already match the target shape $S'$. Therefore, we focus on segments in $C'_{r-1}$ that mismatch $S'$.

To locate those segments, we need a segment-level metric that estimates each segment's contribution to the target shape. Recent work on text-to-image diffusion [Rombach et al., 2022] shows that cross-attention maps expose the correspondence between words and the pixels they describe. We observe an analogous effect in our P2S model: the cross-attention maps indicate which parametric sequence segments are responsible for which geometric parts. As Figure 3 illustrates, when the shape-sequence pair $(S_r, C_r)$ comes from the same CAD model, attention peaks align tightly—e.g., the "Line" segment in $C_r$ attends to the line in $S_r$. Conversely, when $(S', C_r)$ comes from different models, segments irrelevant to $S'$ (such as "Arc") receive low attention scores, while relevant ones (e.g., "Line", "Circle") still correlate well.

Motivated by this insight, we take the cross-attention score between the $i$-th segment of $C'_{r-1}$ and the latent representation of $S'$ as a raw contribution score, denoted as $\mathcal{M}(C'_{r-1}(i), S')$. As absolute magnitudes of $\mathcal{M}$ vary across segments (e.g., [SOL] tends to receive a lower score compared to other segments in Figure 3), we convert this raw value into a scale-invariant relative score:

$$J(i) = |\mathcal{M}(C'_{r-1}(i), S') - \mathcal{M}(C'_{r-1}(i), S'_{r-1})|. \tag{2}$$

$J(i)$ measures how much each segment's influence changes between $S'_{r-1}$ and $S'$. At each iteration, we rank segments by $J(i)$ and mask the $K$ largest ones (in practice, those above the mean $\bar{J}$) to obtain $C^{\text{mask}}_r$. This strategy concentrates computational effort on the segments most responsible for the discrepancy with the target shape $S'$, yielding more efficient and accurate edits.

**Generation.** Starting from the masked parametric sequence $C^{\text{mask}}_r$ from the planning stage, the generation stage calls the MPP model $N$ times to produce a set of candidate parametric sequence $C^1_r, \ldots, C^N_r$. For each candidate $C^n_r$, the MPP model infills the masked segments autoregressively,

Table 1: Quantitative results. **IoU**, **mean CD**, and **median CD** quantify how closely the shape rendered from the edited sequence matches the target geometry. **Edit Dist.** measures how much the edited sequence diverges from the original sequence. **IR** is the percentage of edited sequences that cannot be rendered into a valid shape, and **JSD** is the distributional gap between the generated and target shapes. **Human Eval.** reports the average rank assigned by human annotators.

| Method | IoU↑ | mean CD↓ | median CD↓ | JSD↓ | IR (%)↓ | Edit Dist.↓ | Human Eval.↓ |
|---|---|---|---|---|---|---|---|
| GPT-4o | 0.247 | 0.107 | 0.0171 | 0.737 | 25.1 | 21.12 | 4.57 |
| o4-mini | 0.185 | 0.118 | 0.0283 | 0.748 | 32.95 | 22.49 | 5.40 |
| o4-mini-high | 0.193 | 0.100 | 0.0200 | 0.745 | 40.5 | 25.27 | 5.37 |
| CAD-Diffuser | 0.548 | 0.097 | 0.0093 | 0.689 | 5.7 | 17.29 | 1.94 |
| FlexCAD | 0.447 | 0.029 | 0.0065 | 0.634 | 15.3 | 22.29 | 2.35 |
| **Ours** | **0.687** | **0.009** | **0.0031** | **0.621** | **3.1** | **16.87** | **1.37** |

drawing on the CAD knowledge it gained through pre-training and task-specific fine-tuning:

$$P(C_r^n \mid C_r^{\text{mask}}) = \prod_{t=1}^{T} P(C_r^{n,t} \mid C_r^{\text{mask}}, C_r^{n,<t}), \qquad (3)$$

where $C_r^{n,t}$ is the token autoregressively generated at step $t$ and $C_r^{n,<t}$ denotes the partial sequence generated so far.

**Verification.** It receives candidate parametric sequences $C_r^1, \ldots, C_r^N$ produced in the generation stage, and selects the one whose corresponding shape latent representation lies closest to that of the target shape $S'$. As the P2S model builds a bridge between parametric sequences and the latent space of geometric shapes, we diffuse every candidate sequence and encode the target shape into the shape latent space and measure their Euclidean distance. For a candidate sequence $C_r^n$, we obtain its latent vector through a reverse process of the P2S model, denoted as $\mathcal{F}(C_r^n)$. The target shape $S'$ is embedded by the P2S model's shape encoder $E_s$, denoted as $E_s(S')$. We then choose:

$$C_r' = \arg \min_{\tilde{C} \in \mathcal{Q}} \|\mathcal{F}(\tilde{C}) - E_s(S')\|_2, \qquad (4)$$

where $\mathcal{Q}$ is a priority queue that retains the $X$ best candidate sequences seen up to iteration $r$. Maintaining this cross-iteration priority queue enlarges the search horizon: it rescues high-quality candidates generated in earlier rounds and attenuates the impact of occasional noisy candidates, leading to more reliable convergence toward the target shape.

## 4 Experiment

### 4.1 Experimental Setup

**Datasets.** We train both the P2S and MPP model on the DeepCAD corpus [Wu et al., 2021], which contains about 130k CAD models after removing non-renderable shapes. We keep the official train/validation/test splits. Parametric construction sequences follow the format of Ma et al. [2024]; voxelised SDFs are obtained with `PythonOCC` [Paviot and Contributors, 2025], `Trimesh` [Dawson-Haggerty and Contributors, 2025], and `meshtosdf` [Kleineberg, 2025]. For evaluation, we adopt the 2k test set from CAD-Editor [Yuan et al., 2025]. Each data provides an "edited" parametric sequence whose rendered voxelized SDFs serves as the target shape in our task, while the the accompanying textual instructions are discarded. Following common practice [Yuan et al., 2025, Wang et al., 2025c, Cheng et al., 2023], we generate 5 outputs for each test case for fair comparison and utilize truncated SDFs with the distance range in $[-0.2, 0.2]$.

**Implementation Details.** The MPP model is finetuned from Llama3-8b-Instruct [Meta AI, 2024] with a LoRA [Hu et al., 2022] rank of 32 under the batch of 16 for 60 epochs on 8 A100-40GB-SXM GPUs. The initial learning rate is set to $5e-4$ with maximal token length of 1024. The P2S model is trained on the same hardware with a total batch size of 8 and an initial learning rate of $5e-5$ for 600k steps. The maximum iteration of the plan-generation-verify framework is set as 10.

**Metrics.** As the first work for geometry-driven parametric editing, we introduce evaluation metrics that assess the task from the following perspectives. 1) *Target shape adherence.* The shape rendered

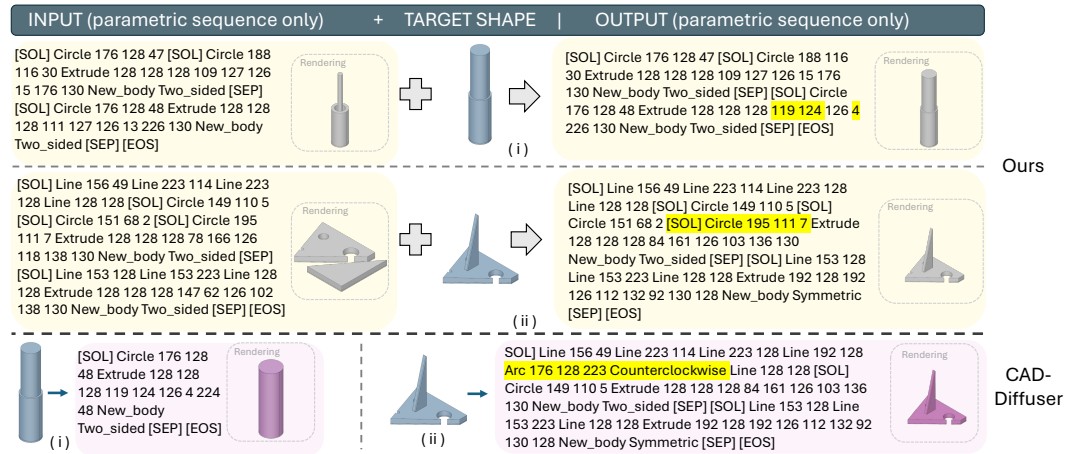

Figure 5: Comparison of edited parametric sequences. Our method is compared with the strongest baseline, CAD-Diffuser. Yellow highlights mark segments where our method preserves the original sequence with only minimal changes, whereas CAD-Diffuser introduces larger modifications.

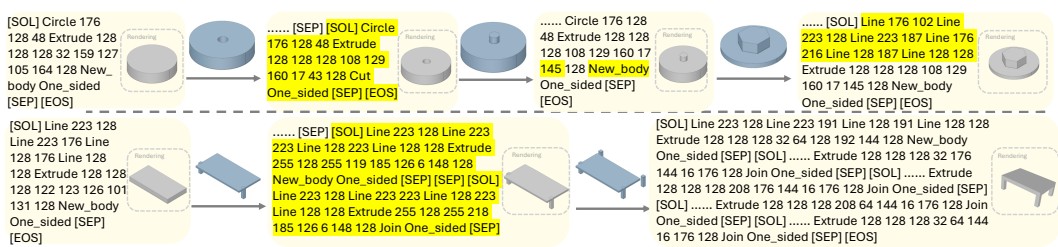

Figure 6: Iterative editing capability of CADMorph. Yellow highlights indicate the modified segments, while "......" denotes parts left unchanged from the previous parametric sequence.

from the updated parametric sequence should closely reproduce the target shape. We adopt **Chamfer Distance (CD)** [Wu et al., 2021] and **Intersection over Union (IOU)** [Ma et al., 2024] to quantify the surface and volume consistency between the result shape and the target shape. 2) *Source sequence consistency.* The updated parametric sequence should be loyal to the original sequence, instead of generating a completely different sequence. We measure this with the **Edit Distance (Edit Dist.)** [Ma et al., 2024] between the result sequence and the original parametric sequence. 3) *Generation Fidelity.* These metrics reflect the overall quality of the outputs. We report the **Invalid Rate (IR)** [Wu et al., 2021], which is the fraction of updated parametric sequences that fail to render into a valid shape, and **Jensen-Shannon Divergence (JSD)** [Wu et al., 2021] between the distribution of generated outcomes and the ground-truth with respect to rendered point clouds.

## 4.2 Comparison with Existing Methods

**Baseline Methods.** We benchmark CADMorph against three families of baselines. 1) Vision-language models (VLMs) their reasoning variants. We test **GPT-4o**-241210, **o4-mini**-250421, **o4-mini-high**-250421. Following standard practice [Khan et al., 2024b, Wu et al., 2024], each model is provided with a explanation of the CAD construction sequence grammar and multi-view renderings of the target shape. 2) Reverse engineering methods. We adopt **CAD-Diffuser** [Ma et al., 2024], which is a reproducible state-of-the-art method that reconstructs parametric sequence directly from geometry. 3) Classical editing methods. We adopt **FlexCAD** [Zhang et al., 2024], which edits a given parametric sequence without explicit guidance. Reverse-engineering methods ignore the user intent embedded in the original sequence, while classical editing methods disregard the visual cues in the target geometry; neither is designed for geometry-driven parametric editing. We include them to highlight the difficulty of the task and demonstrate the advantage of our method.

Table 2: Quantitative results for ablation studies.

| No. | Method | IoU↑ | mean CD↓ | JSD↓ | IR (%)↓ | Edit Dist. ↓ |
|-----|--------|------|----------|------|---------|--------------|
| **A.** | **Ours** | **0.687** | **0.009** | **0.621** | **3.1** | **16.87** |
| B. | - Candidate queue $\mathcal{Q}$ of verification stage | 0.619 | 0.010 | 0.635 | 3.3 | 16.93 |
| C. | - Verification stage | 0.517 | 0.023 | 0.633 | 10.7 | 18.42 |
| D. | - Planning stage | 0.447 | 0.029 | 0.634 | 15.3 | 22.29 |

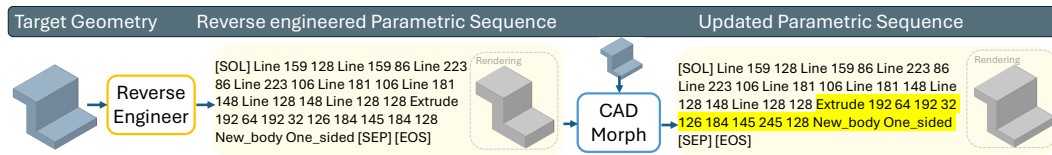

Figure 7: Use CADMorph after conventional reverse engineering to further improve shape fidelity.

**Quantitative Results.** Table 1 shows results. Our method achieves better performance on all metrics. VLM (GPT-4o) struggles to output syntactically valid sequences (high IR) and consequently produces shapes with poor quality (low IoU, high CD and JSD). Adding visual-reasoning enhancements (o4-mini and o4-mini-high) does not yield gains, underscoring the gap between generic VLM capabilities and the demands of geometry-driven parametric editing. Traditional pipelines perform better: CAD-Diffuser and FlexCAD outperform the VLMs, yet both still fall short of simultaneously preserving the structure of the original construction sequence and reproducing the target shape.

**Qualitative Results.** Figure 4 shows results. Our method consistently yields shapes that most closely reproduce the target shape. GPT-4o and its reasoning variants frequently generate invalid sequences, and their valid shape renderings diverge from the target shape. FlexCAD almost always produces syntactically valid sequence, but as it lacks visual guidance, its results rarely resemble the target shape. CAD-Diffuser performs better than GPT-4o and FlexCAD, yet it still falls short of our method.

Figure 5 compares the edited parametric sequences from our method and the strongest baseline, CAD-Diffuser. Whereas our approach strives for the smallest possible edits needed to reproduce the target geometry, CAD-Diffuser often rewrites larger portions of the sequence. In example (i), our method reuses an existing cylinder and simply adjusts its parameters; CAD-Diffuser collapses the two original cylinders into a single one. In example (ii), our method preserves the circular hole from the original sequence, while CAD-Diffuser proposes an adjacent arc. This difference is crucial in practice: drilling a hole and cutting an inward arc involve distinct manufacturing processes. By retaining the hole rather than proposing an arc, our edits leave the downstream manufacturing workflow intact.

**Human Evaluation.** 5 human annotators ranked 200 outputs from each method, jointly evaluating (1) how closely the rendered shape matches the target and (2) how well the edited sequence preserves the original structure. As shown in Table 1, our method receives the highest preference scores.

## 4.3 Downstream Applications

**Iterative Editing Capability.** Figure 6 illustrates CADMorph's iterative editing workflow. In the first round, the user provides an original parametric sequence and a target shape. CADMorph returns an updated sequence that reproduces the target shape. In the subsequent rounds, the output from the previous round is treated as the new "original" sequence. Given a new target shape, CADMorph applies further edits to produce a sequence that meets the new target shape. This loop enables successive refinements while preserving the edit history at each round.

A noteworthy pattern appears in the bottom example of Figure 6 and fifth example of Figure 4: when the given shape is inaccurate—such as legs are not fully flush with the panel—CADMorph silently correct them. We hypothesize that this behavior originates from the MPP model, which absorbed extensive real-world and design knowledge during pre-training and CAD-specific fine-tuning.

**Refining Reverse Engineering Results.** As illustrated in Figure 7, a conventional reverse-engineering pipeline first reconstructs a parametric sequence from the target geometry. CADMorph then takes

this preliminary sequence together with the target shape and outputs a refined sequence that more faithfully reproduces the design—for example, by extending the extrusion height.

### 4.4 Ablation Studies

Table 2 shows ablation studies. 1) Remove the verifier's priority queue (variant **B**). Without the queue, the verifier can only pick the best candidate from the current round rather than from all previous rounds, causing a large IoU decline. This confirms that the priority queue is helpful for retaining high-quality candidates across iterations. 2) Remove verification stage (variant **C**). With no feedback on shape similarity, this variant chooses the candidate at random. 3) Remove planning stage (variant **D**). Segments requiring edits are now chosen at random. Variants C and D leads to a broad drop in performance, highlighting the value of our plan–generate–verify framework.

## 5  Conclusion

We tackle geometry-driven parametric editing: given an original CAD sequence and a target shape, produce an updated sequence that reproduces that shape. To solve this, we introduce CADMorph, an iterative plan–generate–verify framework that couples two complementary components—a Parameter-to-Shape (P2S) diffusion network and an LLM-based Masked Parameter Prediction (MPP) model. Experiments demonstrate that CADMorph surpasses all baselines, and we showcase practical applications, i.e., iterative editing and refining reverse-engineered results. In the future, We will pursue stronger P2S and MPP backbones to boost CADMorph's performance. Besides, we plan to deploy CADMorph as a data-generation engine to synthesize triplets for training a fully end-to-end model.

**Limitations.** 1) Inference latency. Like other test-time-scaling methods that employ verifiers, CADMorph must execute several plan–generate–verify iterations, incurring noticeable runtime. Future work could reduce this overhead by accelerating the component models and parallelizing generation and verification. 2) Test set. Our experiments rely on the CAD-Editor's test set [Yuan et al., 2025], the only publicly available benchmark suited to CADMorph. However, its CAD models are simpler than real-world assemblies, highlighting the need for a richer and more challenging dataset.

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
