# OpenReview forum: "CADMorph: Geometry‑Driven Parametric CAD Editing via a Plan–Generate–Verify Loop"
_NeurIPS.cc/2025/Conference — NeurIPS 2025 poster_

### Official Review · Reviewer_qRBB · 2025-07-02

**Clarity:** 3
**Significance:** 2
**Originality:** 2
**Rating:** 4
**Confidence:** 4

**Summary:**

Given a CAD sequence and a target shape, the proposed method can learn the desired part of the sequence that needs to be changed and generate a new sequence that make the reconstructed shape similar to the target shape. The proposed method has three stages, the planning stage to find the desired part sequence by cross attention map, the generation stage to generate various candidates from the masked sequence, and the verification stage to measure the similarity in the latent space.The experiment results show that the proposed method can efficiently edit the input CAD sequence to make the generated shape similar to the target shape.

**Questions:**

I would like to see response about my concerns raised in the weakness section.

It would be better to provide more challenging examples and studies for the iteration times.

**Ethical Concerns:**

["NO or VERY MINOR ethics concerns only"]

**Final Justification:**

I thank the authors's response.

I hope the authors could add the integration of other input modalities and the challenging cases in their revision as promised

**Limitations:**

yes

**Quality:**

3

**Strengths And Weaknesses:**

Strengths:

1. The cross-attention map is the most interesting part. The edited shape driven by the target shape successfully acquires more similar features to the target shape.

2. The experimental results look promising.

3. The paper is well-structured and easy to follow.

Weaknesses:

1. The novelty and contribution are limited. The key modules are directly adopted from existing works. The main contribution is the plan-generation-verification pipeline and the cross-attention map between the sequence and the shape, which is a combination of existing modules.

2. There are concerns about the practical usage of the method. The method requires a CAD shape program and a 3D CAD model as input. When designers are editing the CAD program, it is less practical to find a target shape in real-time. If the target shape could be an image or sketch, it would likely have more practical applications.

3. From the results in the paper and supplementary materials, the target shape often appears close to the source shape in many parts. It would be beneficial to see more challenging cases and significant topology editing, such as adding numerous holes. Current examples seem naive, and the editing does not appear significant. Along with the inference latency, the practical usage is limited, raising questions about the community's potential interest in this paper.

4. An additional experiment showing the accuracy and time effects of different maximum iteration times would be beneficial.

---

> ### Author Rebuttal · Authors · 2025-07-30
>
> We sincerely thank the reviewer for the valuable feedback. We address the concerns as follows:
>
> ### Novelty and contribution
> The essence of CADMorph lies in utilizing domain-specific foundation models, namely the pretrained priors on domain data, to perform tasks they were not explicitly trained for, at test time, where data scarcity hinders training-based solutions. This is achieved by leveraging existing modules, namely LLM-based MPP and LDM-based P2S, as the foundation, and orchestrating them through a plan-generation-verify pipeline. This pipeline synergistically triggers the sequence editing capability guided by geometry, enabling capabilities beyond what either existing MPP or P2S modules were originally designed to perform.
>
> We position our contribution as developing simple yet effective strategies that empower existing foundation models to gain new capabilities at test time, for what the era of foundation models calls in the face of ever-complicating tasks where data is too hard to collect. To help clarify the significance of our approach, we highlight two influential works that share similar philosophy. Although they differ in domain and specific strategies, they exemplify the value of enabling existing modules to go beyond their original purposes without retraining.
> _Eg 1 Chain-of-Thought Prompting (arxiv:2201.11903)_ leaves existing LLMs untouched and instead provides demonstrations with intermediate steps at test time. It marks a significant breakthrough in eliciting reasoning capabilities from LLMs.
> _Eg 2 DreamFusion (arxiv:2209.14988)_ keeps the existing Stable Diffusion unchanged and uses it as a prior to optimize a NeRF. It represents a significant step forward in leveraging 2D generation models to enhance 3D generation.
>
> ### Image or sketch could be more practical
> While our work uses SDF as the target shape representation, this does not imply that SDFs must be provided directly in practical applications. In real-world scenarios, we treat the SDF as an interface and convert more accessible input formats, such as images or sketches, into SDFs with reasonable effort. For example, this transformation can be achieved using existing foundations, such as Hunyuan3D-2.5 (arxiv:2506.16504, image-to-mesh, followed by tSDF extraction via marching cube), Magic3DSketch (arxiv:2407.19225, sketch-to-mesh and extract the tSDF via marching cube), or LASDiffusion (arxiv:2305.04461, direct sketch-to-SDF).
> We sincerely thank you for the insightful comments w.r.t. practical usage. We will include the integration of other input modalities in final version. For further details on why we chose SDFs as the interface, please refer to supplementary material Part B.
> ### Challenging cases
> As including images is banned, we provide examples in the form of construction sequences at the end of this rebuttal. The three examples successfully 1) added a stack of holes of equal intervals along a corner-like board, 2) transformed a torus into a rudder-like object, and 3) added fin-like heat sinks onto a flat motherboard.
> ### Accuracy and time effect
> For overall runtime, we include details in supplementary material Part E and Table 1. It is on par with test-time reasoning models (e.g., ChatGPT o3 and o4), indicating that the runtime of CADMorph fails within an acceptable range.
> For effect of maximum number of iterations, we provide an ablation study in response 3c to reviewer t3cp. The results show that the performance gradually saturates as iteration number increases. Qualitative results in supplementary material Part G and Figure 4 (page 7) provide more intuitive examples. Simpler cases (top) require fewer iterations, while more complex ones (bottom) demand more. In future work, we plan to explore a dynamic stopping mechanism to better balance accuracy and runtime based on task complexity.
> Overall, thanks for your valuable feedback. It has helped us refine our work and broaden its appeal. We will revise the paper in accordance with your suggestions.
>
> P.S. Additional challenging cases.
>
> 1. Adding multiple cylinders around the edge of a corner-like board.
>
> Original Sequence:
> ```
> [SOL] Arc 130 126 64 Counterclockwise Line 160 126 Line 221 126 Arc 223 128 64 Counterclockwise Line 223 156 Arc 221 158 64 Counterclockwise Arc 160 219 64 Clockwise Arc 158 221 64 Counterclockwise Line 130 221 Arc 128 219 64 Counterclockwise Line 128 128
> ```
>
> The target shape in the form of construction sequence:
> ```
> [SOL] Arc 130 126 64 Counterclockwise Line 160 126 Line 221 126 Arc 223 128 64 Counterclockwise Line 223 156 Arc 221 158 64 Counterclockwise Arc 160 219 64 Clockwise Arc 158 221 64 Counterclockwise Line 130 221 Arc 128 219 64 Counterclockwise Line 128 128 [SOL] Circle 144 142 4 [SOL] Circle 144 158 4 [SOL] Circle 144 174 4 [SOL] Circle 144 190 4 [SOL] Circle 144 205 4 [SOL] Circle 160 142 4  [SOL] Circle 176 142 4 [SOL] Circle 192 142 4 [SOL] Circle 207 142 4 Extrude 192 64 192 80 128 83 144 134 128 New_body One_sided [SEP] [EOS]
> ```
>
> The edited construction sequence:
> ```
> [SOL] Arc 130 126 64 Counterclockwise Line 160 126 Line 221 126 Arc 223 128 64 Counterclockwise Line 223 156 Arc 221 158 64 Counterclockwise Arc 160 219 64 Clockwise Arc 158 221 64 Counterclockwise Line 130 221 Arc 128 219 64 Counterclockwise Line 128 128 [SOL] Circle 144 142 4 [SOL] Circle 160 142 4  [SOL] Circle 176 142 4 [SOL] Circle 192 142 4 [SOL] Circle 144 158 4 [SOL] Circle 144 174 4 [SOL] Circle 144 190 4 [SOL] Circle 144 205 4 [SOL] Circle 207 142 4 Extrude 192 64 192 80 128 83 144 134 128 New_body One_sided [SEP] [EOS]
> ```
>
> 2. Transforming a torus into a rudder-like object.
>
> Original Sequence:
> ```
> [SOL] Circle 176 128 48 [SOL] Circle 176 128 20 Extrude 128 128 128 53 128 128 151 139 128 New_body One_sided [SEP] [EOS]
> ```
>
> The target shape in the form of construction sequence:
> ```
> [SOL] Circle 176 128 48 [SOL] Circle 176 128 20 Extrude 128 128 128 53 128 128 151 139 128 New_body One_sided [SEP] [SOL] Arc 223 128 12 Clockwise Line 223 223 Line 128 223 Line 128 128 Extrude 128 128 128 117 202 128 22 139 128 Join One_sided [SEP] [SOL] Arc 163 68 12 Clockwise Line 223 103 Line 188 163 Line 128 128 Extrude 128 128 128 187 175 128 29 139 128 Join One_sided [SEP] [SOL] Line 188 93 Line 223 153 Line 163 188 Arc 128 128 12 Clockwise Extrude 128 128 128 187 81 128 29 139 128 Join One_sided [SEP] [SOL] Line 223 128 Line 223 223 Arc 128 223 12 Clockwise Line 128 128 Extrude 128 128 128 117 32 128 22 139 128 Join One_sided [SEP] [SOL] Line 163 68 Line 223 103 Arc 188 163 12 Clockwise Line 128 128 Extrude 128 128 128 39 89 128 29 139 128 Join One_sided [SEP] [SOL] Line 188 93 Arc 223 153 12 Clockwise Line 163 188 Line 128 128 Extrude 128 128 128 39 167 128 29 139 128 Join One_sided [SEP] [EOS]
> ```
> The edited construction sequence:
> ```
> [SOL] Circle 176 128 48 [SOL] Circle 176 128 20 Extrude 128 128 128 53 128 128 151 139 128 New_body One_sided [SEP] [SOL] Line 188 93 Arc 223 153 12 Clockwise Line 163 188 Line 128 128 Extrude 128 128 128 39 167 128 29 139 128 Join One_sided [SEP]  [SOL] Arc 223 128 12 Clockwise Line 223 223 Line 128 223 Line 128 128 Extrude 128 128 128 117 202 128 22 139 128 Join One_sided [SEP] [SOL] Arc 163 68 12 Clockwise Line 223 103 Line 188 163 Line 128 128 Extrude 128 128 128 187 175 128 29 139 128 Join One_sided [SEP] [SOL] Line 188 93 Line 223 153 Line 163 188 Arc 128 128 12 Clockwise Extrude 128 128 128 187 81 128 29 139 128 Join One_sided [SEP] [SOL] Line 223 128 Line 223 223 Arc 128 223 12 Clockwise Line 128 128 Extrude 128 128 128 117 32 128 22 139 128 Join One_sided [SEP] [SOL] Line 163 68 Line 223 103 Arc 188 163 12 Clockwise Line 128 128 Extrude 128 128 128 39 89 128 29 139 128 Join One_sided [SEP] [EOS]
> ```
> 3. Adding multiple fin-like heat sinks onto a mother board.
>
> Original Sequence:
> ```
> [SOL] Line 223 128 Line 223 160 Line 128 160 Line 128 128 Extrude 128 128 128 32 96 128 192 137 128 New_body One_sided [SEP] [EOS]
> ```
> The target shape in the form of construction sequence:
> ```
> [SOL] Line 223 128 Line 223 160 Line 128 160 Line 128 128 Extrude 128 128 128 32 96 128 192 137 128 New_body One_sided [SEP] [SOL] Line 138 128 Line 138 223 Line 128 223 Line 128 128 Extrude 128 128 128 106 96 137 64 171 128 Join One_sided [SEP] [SOL] Line 138 128 Line 138 223 Line 128 223 Line 128 128 Extrude 128 128 128 144 96 137 64 171 128 Join One_sided [SEP] [SOL] Line 138 128 Line 138 223 Line 128 223 Line 128 128 Extrude 128 128 128 78 96 137 64 171 128 Join One_sided [SEP] [SOL] Line 138 128 Line 138 223 Line 128 223 Line 128 128 Extrude 128 128 128 40 96 137 64 171 128 Join One_sided [SEP] [SOL] Line 138 128 Line 138 223 Line 128 223 Line 128 128 Extrude 128 128 128 172 96 137 64 171 128 Join One_sided [SEP] [SOL] Line 138 128 Line 138 223 Line 128 223 Line 128 128 Extrude 128 128 128 211 96 137 64 171 128 Join One_sided [SEP] [EOS]
> ```
> The edited construction sequence:
> ```
> [SOL] Line 223 128 Line 223 160 Line 128 160 Line 128 128 Extrude 128 128 128 32 96 128 192 137 128 New_body One_sided [SEP] [SOL] Line 138 128 Line 138 223 Line 128 223 Line 128 128 Extrude 128 128 128 172 96 137 64 171 128 Join One_sided [SEP] [SOL] Line 138 128 Line 138 223 Line 128 223 Line 128 128 Extrude 128 128 128 211 96 137 64 171 128 Join One_sided [SEP] [SOL] Line 138 128 Line 138 223 Line 128 223 Line 128 128 Extrude 128 128 128 78 96 137 64 171 128 Join One_sided [SEP] [SOL] Line 138 128 Line 138 223 Line 128 223 Line 128 128 Extrude 128 128 128 40 96 137 64 171 128 Join One_sided [SEP] [SOL] Line 138 128 Line 138 223 Line 128 223 Line 128 128 Extrude 128 128 128 172 96 137 64 171 128 Join One_sided [SEP] [SOL] Line 138 128 Line 138 223 Line 128 223 Line 128 128 Extrude 128 128 128 211 96 137 64 171 128 Join One_sided [SEP] [SOL] Line 138 128 Line 138 223 Line 128 223 Line 128 128 Extrude 128 128 128 106 96 137 64 171 128 Join One_sided [SEP] [SOL] Line 138 128 Line 138 223 Line 128 223 Line 128 128 Extrude 128 128 128 144 96 137 64 171 128 Join One_sided [SEP] [EOS]
> ```

---

> > ### Comment · Reviewer_qRBB · 2025-08-06
> >
> > "we include details in supplementary material Part E and Table 1. It is on par with test-time reasoning models (e.g., ChatGPT o3 and o4), indicating that the runtime of CADMorph fails within an acceptable range. For effect of maximum number of iterations, we provide an ablation study in response 3c to reviewer t3cp."
> >
> > 7.26 mins is a little longer for each inference, could you please also add time in the ablation study mentioned above? How is the time change with the iteration increasing?

---

> ### Author Response · Authors · 2025-08-08
> **About the runtime under different number of iterations**
>
> Thanks a lot for your question. Your question has really let us re-assess the inference time of CADMorph under different number of iterations and has insipred us to further optimize the inference spped of our model.
>
> We tested the running time of our model under different number of iterations, as is shown in the following table. The last but one column is the total elapsed time of a single edit. The last column is the edit runtime with model loading overhead removed. As shown in the table, the running time scales in an approximately linear manner if the model loading overhead has been removed.
>
> | Num. of Iterations | IoU (↑) | Mean Chamfer Distance (↓) | JSD (↓) | Invalid Rate (↓) | Edit Distance (↓) | Time Elapsed (minutes, ↓) | Time excluding model loading (minutes, ↓) |
> |---|---|---|---|---|---|---|---|
> | 1  | 0.512 | 0.024 | 0.629 | 10.8% | 19.48 | 1.93 | 0.88 |
> | 2  | 0.634 | 0.015 | 0.627 | 6.8%  | 17.29 | 2.49 | 1.44 |
> | 5  | 0.665 | 0.011 | 0.626 | 4.7%  | 17.31 | 3.88 | 2.83 |
> | 10 (Our Original Setting) | 0.687 | 0.009 | 0.621 | 3.1% | 16.87 | 7.26 | 6.21 |
>
> From the table we can conclude the following advantages under the condition of serving the model online (i.e. it is not reloaded for each inference).
> 1. Compared with GPT 4o, o4 and o4-mini-high, our model surpasses the performance of all the methods at iteration=1, with only 0.88 minutes required, even less than GPT-4o.
> 2. Compared with previous methods like FlexCAD and CAD-Diffuser, the performance of our model is on par with them even at iteration=1. While CADMorph uses a little bit longer time, 0.88 min is tolerable for human users in the LLM era. However, when the time budget increases, the generation performance of our method has a significant increase and excels with previous methods by a large margin, thus showing the potential of test-time scaling which is lacking in previous methods.
>
> Separately, we carefully reviewed the inference speed optimization literature, and has discovered the following ways to accelerate CADMorph with little or no performance drop.
> 1. As is mentioned earlier, we can load the model as a service so as to allieviate extra loading time, which can save ~10% of the time cost.
> 2. For the MPP model, we can adopt the prevailing LLM inference accleration frameworks like Flash Attention 3 (arXiv: 2407.08608) and Flash Attention 2 (arxiv:2307.08691). As MPP model takes up ~30% of the inference time in each round, this may lead to a total speedup of 10% copared with non-optimized version.
> 3. For the P2S model, as it is a latent diffusion model operating on text-based construction sequences and voxelized truncated SDFs, we can also adopt Flash Attention, as well as other speedup tricks like FlashConv (arXiv: 2407.08608) etc. This may also bring about ~1.5x speedup, which accounts for ~70%*(1/1.5)=23% in total. Moreover, storing the SDFs in SSDs or even memories can also bring about substantial speedup depending on hardware settings.
>
> In all, the first version of CADMoprh still has a lot of space to improve its efficiency, with a conservative estimate of ~40% speedup on only engineering efforts conducted without algorithm changes. This shows great potential of improving the efficiency of our algorithm.
>
> Once again, thank you for your question about the detailed inference time in the extra ablation study. This really helps us improve the efficiency of our algorithm!

---

### Official Review · Reviewer_t7b3 · 2025-07-03

**Clarity:** 3
**Significance:** 3
**Originality:** 3
**Rating:** 5
**Confidence:** 3

**Summary:**

This paper tackles geometry-driven CAD editing by proposing a framework that updates a construction sequence to match a target geometry. It combines a parameter-to-shape diffusion model with a masked sequence predictor, forming a plan–generate–verify loop. The method avoids the need for triplet supervision and effectively handles limited editing data.

**Questions:**

1. The qualitative results mainly focus on mechanical parts. I wonder how well the method performs on more complex real-world data, such as the furniture dataset from BrepGen [1]. For example, could the authors try morphing between two different tables? That would be an interesting demonstration.
2. What is the inference time of CADMorph compared to other baselines? It would be helpful to understand the practical cost of the iterative loop.
3. Can the authors include some failure cases and discuss typical failure modes?

[1] Xu, X., Lambourne, J., Jayaraman, P., Wang, Z., Willis, K., & Furukawa, Y. (2024). Brepgen: A b-rep generative diffusion model with structured latent geometry. ACM Transactions on Graphics (TOG), 43(4), 1-14.

**Ethical Concerns:**

["NO or VERY MINOR ethics concerns only"]

**Final Justification:**

My original concern was that this method might not work well on complex data, such as furniture. The author has addressed this concern. I believe the proposed plan–generate–verify framework, which does not require ground truth editing data, is novel.

**Limitations:**

Yes

**Paper Formatting Concerns:**

No.

**Quality:**

3

**Strengths And Weaknesses:**

**Strengths:**
1. The proposed plan–generate–verify framework is well-motivated. Traditional CAD editing methods require triplet supervision ⟨original sequence, target geometry, updated sequence⟩, where the updated sequence is especially difficult to obtain. This framework effectively removes the need for such supervision.
2. Leveraging cross-attention maps to localize editable segments is an interesting and intuitive idea.
3. The paper is well-written and easy to follow.

**Weaknesses:**

While the proposed method performs well on benchmark examples, its applicability to real-world scenarios remains unclear. The generation process relies entirely on the MPP model to fill in masked segments based on limited context. In complex CAD applications, such as furniture design, where the geometric difference is large and structural edits are nontrivial, the required updates may involve high-level understanding of the geometry. This can make it difficult for the model to converge or produce valid outputs, even after many iterations.

---

> ### Author Rebuttal · Authors · 2025-07-30
>
> We sincerely thank the reviewer for his positive feedback. We feel delighted that the reviewer recognizes the training-free framework design, as well as the editing segmentation localization via cross-attention maps.
>
> The concerns of the reviewer mainly fall into three categories. Here we will address them in detail.
>
> 1.	Applicability in scenarios with large difference. CADMorph is primarily designed for editing tasks with medium-level differences between the source and the target. In cases where the difference is large, a more suitable approach would be to first apply reverse engineering to obtain the construction sequence of the target shape,  and then use CADMorph to iteratively refine this sequence and  improve shape fidelity, as shown in the downstream application part in this paper (Section 4.3). Besides, directly editing CAD models with large geometric discrepancies is one of the major sources of failures, as is shown in Part H and Figure 1 in the supplementary materials.
>
> 2.	Applicability in real-world scenarios like furniture. We manually selected a pair of tables from DeepCAD dataset, where the source is a coffee table with legs positioned at the corners, and the target is a long and thin table with its legs tall and stretched in. Due to the restriction of using images during the rebuttal phase, we exhibit this editing example in the form of construction sequences at the end of this rebuttal. Additionally, a similar example can be found in the example 5 of Figure 4 in the main paper, with analysis provided in lines 289-292.
>
>     For Brepgen mentioned by the reviewer, this dataset is in the form of Breps and lacks the corresponding construction sequence (CSG) that captures the drawing process. Since CADMorph focus on construction sequences, we did not include this dataset in our experiment. For other datasets with similar complexity to Brepgen (e.g., intricately designed furniture) and available in the form of CSGs, we currently did not have access to such data, as discussed in the limitation section.
>
> 3.	Running time. We have included a running time analysis in part E and Table 1 in the supplemental material. The running time of CADMorph is comparable to or shorter than prevailing 3D generation and editing methods like Magic3D (arxiv:2211.10440), Zero-1-to-3 (arxiv:2303.11328), DreamFusion (arxiv:2209.14988), GaussianEditor (arxiv:2311.14521), and GaussCtrl (arxiv:2403.08733), which typically takes tens of minutes or even hours. Moreover, CADMorph’s running time is on par with test-time reasoning models such as ChatGPT o3 and o4. These comparisons indicate that the running time of CADMorph is within an acceptable range.
>
> 4.	Failure cases. This can be found in part H and Figure 1 in the supplementary material. To recap, this shows that CADMorph may stop at somewhere in the middle during editing, and it still does not have the knowledge of the rendering process of the CAD kernel. We consider solving these issues in future work.
>
> In all, we sincerely thank the reviewer for the positive feedback, and we will seriously consider the integration of B-reps in future work.
>
> P.S. Another example of editing tables in DeepCAD dataset.
>
> Original Construction Sequence:
>
> ```
> [SOL] Line 223 128 Line 223 223 Line 128 223 Line 128 128 Extrude 128 128 128 61 75 128 113 134 128 New_body One_sided [SEP] [SOL] Line 223 128 Line 223 223 Line 128 223 Line 128 128 Extrude 255 128 255 61 86 128 11 224 128 Join One_sided [SEP] [SOL] Line 223 128 Line 223 223 Line 128 223 Line 128 128 Extrude 255 128 255 163 86 128 11 224 128 Join One_sided [SEP] [SOL] Line 223 128 Line 223 223 Line 128 223 Line 128 128 Extrude 255 128 255 61 188 128 11 224 128 Join One_sided [SEP] [SOL] Line 223 128 Line 223 223 Line 128 223 Line 128 128 Extrude 255 128 255 163 188 128 11 224 128 Join One_sided [SEP] [EOS]
> ```
>
> The target shape in the form of construction sequence:
>
> ```
> [SOL] Line 176 128 Line 176 223 Line 128 223 Line 128 128 Extrude 128 128 128 83 37 128 182 123 128 New_body One_sided [SEP] [SOL] Line 170 128 Line 170 223 Line 128 223 Line 128 128 [SOL] Line 167 131 Line 167 220 Line 131 220 Line 131 131 Extrude 255 128 255 93 209 123 162 138 128 Join One_sided [SEP] [SOL] Line 223 128 Line 223 223 Line 128 223 Line 128 128 Extrude 255 128 255 98 204 123 15 219 128 Join One_sided [SEP] [SOL] Line 223 128 Line 223 223 Line 128 223 Line 128 128 Extrude 255 128 255 98 67 123 15 219 128 Join One_sided [SEP] [SOL] Line 223 128 Line 223 223 Line 128 223 Line 128 128 Extrude 255 128 255 143 204 123 15 219 128 Join One_sided [SEP] [SOL] Line 223 128 Line 223 223 Line 128 223 Line 128 128 Extrude 255 128 255 143 67 123 15 219 128 Join One_sided [SEP] [EOS]
> ```
>
> The edited construction sequence:
> ```
> [SOL] Line 176 128 Line 176 223 Line 128 223 Line 128 128 Extrude 128 128 128 83 37 128 182 123 128 New_body One_sided [SEP] [SOL] Line 170 128 Line 170 223 Line 128 223 Line 128 128 [SOL] Line 167 131 Line 167 220 Line 131 220 Line 131 131 Extrude 255 128 255 93 209 123 162 138 128 Join One_sided [SEP] [SOL] Line 223 128 Line 223 223 Line 128 223 Line 128 128 Extrude 255 128 255 98 204 123 15 219 128 Join One_sided [SEP] [SOL] Line 223 128 Line 223 223 Line 128 223 Line 128 128 Extrude 255 128 255 98 67 123 15 219 128 Join One_sided [SEP] [SOL] Line 223 128 Line 223 223 Line 128 223 Line 128 128 Extrude 255 128 255 143 204 123 15 219 128 Join One_sided [SEP] [SOL] Line 223 128 Line 223 223 Line 128 223 Line 128 128 Extrude 255 128 255 143 67 123 15 219 128 Join One_sided [SEP] [EOS]
> ```

---

> > ### Comment · Reviewer_t7b3 · 2025-08-03
> > **Discussions**
> >
> > Thank you for your detailed response. I wonder how to visualize the construction sequence that you posted.

---

> > > ### Author Response · Authors · 2025-08-04
> > > **Outline of the visualization method.**
> > >
> > > Thank you for your reply. Due to the strict ban on providing code and visual content, here we can only provide an outline to visualize the text provided. The text is in general a flattened and texturalized version of vectorized construction sequences in DeepCAD [1]. In this sense, the visualization step is as follows.
> > >
> > > 1.  Re-assemble the text back to the original DeepCAD vectors.
> > >
> > >     a. For each word, look up the original encoding as described in the original DeepCAD paper and its open-sourced code. Convert it into the corresponding numeric form.
> > >
> > >     b. For each number, directly put them into the corresponding slots as described in the grammar of DeepCAD.
> > >
> > >     c. For the empty slots, pad them as -1, which is in line with the DeepCAD grammar.
> > >
> > > 2. Use the original code (utils/show.py) in DeepCAD to convert the vector into visual contents like images.
> > >
> > > Once again, thanks a lot for the discussion. Hope these steps may assist you.
> > >
> > > [1] Wu, Rundi, Chang Xiao, and Changxi Zheng. "Deepcad: A deep generative network for computer-aided design models." Proceedings of the IEEE/CVF International Conference on Computer Vision. 2021.

---

> > > > ### Comment · Reviewer_t7b3 · 2025-08-04
> > > >
> > > > Thank you for your reply. My original concern was that this method might not work well on complex data, such as furniture. The author has addressed this concern. I believe the proposed plan–generate–verify framework, which does not require ground truth editing data, is novel. I disagree with reviewer qRBB’s comment that this work has limited novelty and contribution due to the core module being adapted from existing work. I will increase my rating to accept as a show of support for the paper.

---

> > > > > ### Author Response · Authors · 2025-08-05
> > > > > **Thank you for your recognition!**
> > > > >
> > > > > Dear reviewer,
> > > > >
> > > > > We sincerely thank you for your strong recognition to our work. It is a pleasure to discuss with you about the applicability to shape with large differences, as well as extending our work to real-world scenarios. We also sincerely appreciate and admire your steadfast commitment to upholding your opinion throughout the review process.
> > > > >
> > > > > Best regards,
> > > > > Authors of paper 15168

---

### Official Review · Reviewer_ZSgx · 2025-07-03

**Clarity:** 3
**Significance:** 3
**Originality:** 4
**Rating:** 5
**Confidence:** 2

**Summary:**

The paper presents a novel method called CADMorph for geometry-driven parametric CAD editing: given an original CAD parametric sequence from a source object, it generates an updated sequence for a target object that is not too far-off from the original sequence while successfully achieving the target shape.

The method proposes an iterative plan-generate-verify framework by utilizing parameter-to-shape (P2S) model based on a latent diffusion model and masked-parametric-model (MPP) based on a large language model (LMM). P2S model does the planning by locating the segments requiring edits via cross-attention maps and replace them with mask tokens. MPP model infills the mask tokens to propose the edited candidates. The plan and generate steps are repeated iteratively in which verify step is performed in each iteration to select the candidate closest to target shape.

**Questions:**

I am mainly concerned about the robustness of the method. I would appreciate if the authors can provide some insights on the number of candidates, masking proportion, verify iterations to the robustness of the final outputs. I expect a small ablation study for this.

**Ethical Concerns:**

["NO or VERY MINOR ethics concerns only"]

**Final Justification:**

This paper is one of the first to address the task of geometry-driven parametric CAD editing. The proposed method is shown to be effective than using LLMs (GPT 4o, etc.). The additional ablation study looks reasonable.

There is a concern from reviewer qRBB, "There are concerns about the practical usage of the method. The method requires a CAD shape program and a 3D CAD model as input. When designers are editing the CAD program, it is less practical to find a target shape in real-time. If the target shape could be an image or sketch, it would likely have more practical applications.". However, I do not find this to be much an issue due to the recent advances in X-to-3D models, as pointed out in the rebuttal by the authors.

I suggest the authors to include these extra results in the final revision.

**Limitations:**

Yes.

**Paper Formatting Concerns:**

No formatting concern.

**Quality:**

3

**Strengths And Weaknesses:**

I am not too familiar with the parametric representation, dataset, and related works. Based on my understanding, here are some of the strengths and weaknesses.

Strengths:
1. This paper is one of the first to address the task of geometry-driven parametric CAD editing. The challenges of the task is well-explained in the introduction. The proposed method also does not necessitates scarce triple data (original sequence, target geometry, updated sequence).
2. The proposed plan-generate-verify framework works better than LLM prompting and the baseline methods (CAD-diffuser and FlexCAD).

Weaknesses:
1. The methods seems to involve a lot of randomness. The generate step produces a set of N candidates without any constraint (except for the mask tokens). It is not clear from the paper on how many candidates, masking proportion, and number of iterations should be done to ensure generation robustness.
2. The masking in the plan step is based on the cross-attention of the latent representations. For shapes with complex objects, the attention scores might become not accurate.
3. The verification step is also performed to minimize the distance in latent representation. There is no explicit geometry signal on this step. This way, the method can struggle with CAD models with intricate details (e.g., with tiny parts or holes).

---

> ### Author Rebuttal · Authors · 2025-07-30
>
> Thank you for your professional and insightful review—your comments pinpoint exactly the challenges that motivated our design of CADMorph. We appreciate your recognition that our framework is the first to address the task of geometry-driven parametric CAD editing while eliminating the reliance on scarce editing triplets.
>
> Below is our reply to the weaknesses and questions you’ve mentioned.
>
> 1.	About the _robustness_ of our method. Our framework involves several hyperparameters, e.g., number of candidates, masking proportion and number of iterations We conduct additional ablation studies and find that our framework performs robustly across a wide range of those hyperparameters. Specifically, the number of candidates has a minor impact on performance. For the masking proportion and the number of iterations, once they exceed a certain threshold, further increases do not lead to significant performance differences. Below is the detailed explanation of our newly added ablation.
>
>     a)	 For the number of candidates, we ran an additional ablation study about different numbers of candidates proposed at each iteration. The result is shown in the table below. From the result we can observe that the candidate number only slightly affects the result of CADMorph. This is because a cross-iteration priority queue is maintained, which always retains the $X$ best candidates seen up to iteration $r$ (see line 214-217). As a result, CADMorph is less sensitive to the specific choice of the number of candidates at each iteration.
>
>     |Candidate Number  |IoU($\uparrow$)  |Mean Chamfer Distance($\downarrow$)  |JSD($\downarrow$)   |Invalid Rate($\downarrow$) |Edit Distance($\downarrow$)   |
>     |--|--|--|--|--|--|
>     |1  |0.658  |0.012   |0.629   |3.5%  |17.10   |
>     |2  |0.688  |0.009   |0.632   |3.2%  |16.93   |
>     |5 (Our Original Setting) |0.687  |0.009   |0.621   |3.1%  |16.87   |
>     |10 |0.691  |0.013   |0.623   |3.3%  |17.02   |
>
>     b)	For the _masking proportion_, we explore three settings: 1) applying purely random masking without using the proposed latent discrepancy $J$; 2) masking the top 10% tokens with the highest latent discrepancy $J$; 3) masking tokens where the latent discrepancy $J$ exceeds the sequence average $\bar{J}$ (our original setting in the paper). The result is listed in the following table. Although we did not have enough time to investigate more fine-grained settings, the current results still offer valuable insights. First, both the top-10% and above-average masking strategies outperform purely random masking, suggesting that the framework is effective under different masking proportions.Second, the top-10% strategy performs worse than the above-average strategy. We hypothesize this is because masking only the most discrepant tokens may overly constrain the editing process, making it harder to adapt to large geometric differences.
>
>     |Masking Strategy  |IoU ($\uparrow$)  |Mean Chamfer Distance ($\downarrow$)  |JSD ($\downarrow$)   |Invalid Rate ($\downarrow$) |Edit Distance ($\downarrow$)   |
>     |--|--|--|--|--|--|
>     |Purely random masking 50% of primitives without P2S guidance  |0.497  |0.027   |0.633   |14.7%  |22.36   |
>     |Masking top 10% w.r.t. latent discrepancy  |0.643  |0.014   |0.635   |4.8%  |19.47   |
>     |Masking above avg latent discrepancy (Our Strategy) |0.687  |0.009   |0.621   |3.1%  |16.87   |
>
>     c)	For the number of iterations. We set the maximum number of iterations as 10 throughout our experiments. To evaluate its impact, we conduct an ablation study with the maximum number of iterations set to 1, 2, 5 and 10. From the table we can observe that the performance growth tends to saturate as the iterations increases. Additionally, qualitative results are presented in Part G and Figure 4 in the supplementary material. Specifically, simpler editing cases (top example in Fig 4) typically converge in fewer iterations, while more complex editing cases (bottom example) require more iterations.
>
>     |Num. of Iterations  |IoU($\uparrow$)  |Mean Chamfer Distance($\downarrow$)  |JSD($\downarrow$)   |Invalid Rate($\downarrow$) |Edit Distance($\downarrow$)   |
>     |--|--|--|--|--|--|
>     |1  |0.512  |0.024   |0.629   |10.8%  |19.48   |
>     |2  |0.634  |0.015   |0.627   |6.8%  |17.29   |
>     |5  |0.665  |0.011   |0.626   |4.7%  |17.31   |
>     |10 (Our Original Setting) |0.687  |0.009   |0.621   |3.1%  |16.87   |
>
> 2. About complicated objects or objects with intricate details. We discussed a potential solution at the end of part B in the supplementary material. The main idea leverages the extensibility and Boolean algebra support of SDFs, allowing us to decompose a complex CAD model into a set of sketch-extrusion pairs. Each pair maintains a manageable level of complexity and scale, making it easier for cross-attention mechanisms and latent representations to capture the relevant details.  In future work, we plan to further develop this solution and collect complex CAD data to validate its effectiveness. We appreciate your insightful observation and value the opportunity to discuss future directions on this topic.
>
> Once again, we sincerely thank the reviewer for the recognition to our work, as well as the discussion about the future direction of research in this field.

---

> > ### Comment · Reviewer_ZSgx · 2025-08-04
> >
> > I have read the authors' rebuttal. The extra ablation study looks good to me. I have nothing to add.

---

> ### Author Response · Authors · 2025-08-04
> **Thank you for your recognition and insightful discussion!**
>
> Dear reviewer,
>
> We sincerely thank you for your strong recognition to our work, as well as the insightful discussion on the model robustness and applying our model to more complicated objects with more intricate details!
>
> Sincerely,
>
> Authors of paper 15168

---

### Official Review · Reviewer_t3cp · 2025-07-03

**Clarity:** 2
**Significance:** 3
**Originality:** 3
**Rating:** 4
**Confidence:** 3

**Summary:**

This paper introduces CADMorph, a novel approach for geometry-driven parametric CAD editing. The task takes an original parametric CAD construction sequence and a target geometric shape as inputs, and produces an updated parametric sequence that renders the target shape while preserving the structure of the original sequence. The method employs an iterative plan-generate-verify framework utilizing two complementary models: a Parameter-to-Shape (P2S) latent diffusion model that maps parametric sequences to geometric shapes, and a Masked Parameter Prediction (MPP) model based on finetuned LLaMA-3. In each iteration, the planning stage analyzes cross-attention maps from the P2S model to identify segments requiring modification, the generation stage uses the MPP model to propose semantically valid edits for masked segments, and the verification stage selects the candidate sequence whose shape embedding best matches the target. The approach circumvents data scarcity by training the two models separately without requiring triplet supervision data.

**Questions:**

1. Could you provide specific timing comparisons between CADMorph and baseline methods? How many iterations are typically needed for convergence, and what is the wall-clock time for editing a single CAD model?

2. Have you tested CADMorph on more complex industrial CAD models with hundreds of operations? What are the main bottlenecks when scaling to such models?

3. Why was voxelized tSDF chosen over other 3D representations? Have you experimented with mesh-based or implicit representations, and how do they compare?

4. Did you explore other ways to combine the P2S and MPP models? For example, could they be trained jointly or combined in a different framework?

**Ethical Concerns:**

["NO or VERY MINOR ethics concerns only"]

**Final Justification:**

After reading the rebuttal and carefully considering the opinions of other reviewers, I remain positive about this paper. I believe the authors' response has adequately addressed my concerns. I recommend a **Boardline Accept**.

**Limitations:**

yes

**Paper Formatting Concerns:**

No major formatting issues.

**Quality:**

3

**Strengths And Weaknesses:**

### Strengths

1. The paper addresses geometry-driven parametric CAD editing, a practically important but underexplored task that bridges the gap between reverse engineering and traditional CAD editing methods.

2. The use of cross-attention maps from the P2S diffusion model to identify which parametric segments need editing is creative and well-motivated. The visualization in Figure 3 convincingly demonstrates this insight.

3. The approach cleverly avoids the need for scarce triplet training data (original sequence, target shape, updated sequence) by leveraging two separately trained models, making the method practical to implement.

4. The method outperforms strong baselines including GPT-4o and specialized CAD methods on multiple metrics. The human evaluation further confirms its superiority.

### Weaknesses

1. **Limited evaluation dataset**: The experiments rely solely on the CAD-Editor test set with 2k examples, which the authors acknowledge contains simpler models than real-world assemblies. This limits the generalizability claims of the approach.

2. **Computational efficiency concerns**: As a test-time scaling method requiring multiple iterations of plan-generate-verify cycles, the inference latency is a significant limitation for practical deployment. The paper lacks detailed runtime analysis.

3. **Limited ablation studies**: The ablations don't explore important design choices such as the number of iterations, different masking strategies, or the impact of the number of candidates N in the generation stage.

---

> ### Author Rebuttal · Authors · 2025-07-31
>
> We sincerely thank the reviewer for the encouraging and insightful feedback. We are delighted that youour contribution are recognized: (i) the practical importance and relative under exploration of geometry-driven parametric CAD editing, (ii) our use of cross attention maps from the P2S diffusion model as an intuitive, geometry-aware guide for pinpointing edit regions, and (iii) our strategy of leveraging domain-specific foundation models to sidestep the scarcity of triplet supervision and still achieve strong performance.
>
> The weakness proposed by the reviewer mainly fall into the following categories. Here we will address them in detail.
>
> 1.	Dataset limitations. As noted in the limitation section, the ML/AI community currently lacks industry-level complex CAD models in the form of CSG. We hope this limitation will not lead to a negative impression of our submission. In the following, we try our best to show the potential and generalizability of our approach.
>
>     a.	We selected some complicated CAD models in current dataset, namely DeepCAD and performed additional experiments. Due to the restriction of including images during the rebuttal phase, the results are presented in the form of construction sequences at the end of the response to Reviewer qRBB. These examples demonstrate that CADMorph successfully: 1) added numerous holes to a corner-like board, 2) transformed a torus into a rudder-like object, and 3) added a stack of evenly distributed fins onto a rectangle motherboard. These examples suggest promising scaling ability of CADMorph to more complex CAD models.
>
>     b.	To address the challenge posed by the lack of complex CAD models, we discussed a potential solution at the end of part B in the supplementary material. The main idea is to decompose complex CAD models into sketch-extrusion pairs, use CADMorph on them one by one, and finally assemble them via Boolean operations of SDFs. We plan to invest further effort in making this proposal practical and effective in future work.
>
>     c.	Previously accepted work like CAD-Editor (arxiv:2502.03997), FlexCAD (arxiv:2411.05823), CAD-Llama (arxiv:2505.04481), CADFusion (arxiv:2501.19054) and Text2CAD (arxiv:2409.17106)  were also conducted on the same datasets and have shown significant impact in both research and industry. In this sense, we believe that our experiments conducted on this dataset provides a sufficient starting point for unlocking further capabilities of domain-specific foundation models in the CAD domain.
>
> 2.	Running time. We have included a running time analysis in part E and Table 1 in the supplemental material. The running time of CADMorph is comparable to or shorter than prevailing 3D generation and editing methods like Magic3D (arxiv:2211.10440), Zero-1-to-3 (arxiv:2303.11328), DreamFusion (arxiv:2209.14988), GaussianEditor (arxiv:2311.14521), and GaussCtrl (arxiv:2403.08733), which typically takes tens of minutes or even hours. Moreover, CADMorph’s running time is on par with test-time reasoning models such as ChatGPT o3 and o4. These comparisons indicate that the running time of CADMorph is within an acceptable range.
>
> 3.	About additional ablation studies.
>
>     a.	For the number of candidates, we ran an additional ablation study about different numbers of candidates proposed at each iteration. The result is shown in the table below. From the result we can observe that the candidate number only slightly affects the result of CADMorph. This is because is a cross-iteration priority queue is maintained, which always retains the $X$ best candidates seen up to iteration $r$ (see line 214-217). As a result, CADMorph is less sensitive to the specific choice of the number of candidates.
>
>     |Candidate Number  |IoU($\uparrow$)  |Mean Chamfer Distance($\downarrow$)  |JSD($\downarrow$)   |Invalid Rate($\downarrow$) |Edit Distance($\downarrow$)   |
>     |--|--|--|--|--|--|
>     |1  |0.658  |0.012   |0.629   |3.5%  |17.10   |
>     |2  |0.688  |0.009   |0.632   |3.2%  |16.93   |
>     |5 (Our Original Setting) |0.687  |0.009   |0.621   |3.1%  |16.87   |
>     |10 |0.691  |0.013   |0.623   |3.3%  |17.02   |
>
>     b.	For the masking strategy, in current experiment, we mask the parts of the sequence where the discrepancy of latents $J$ exceeds the average $\bar{J}$ across the sequence (see line 199). We add two ablation studies: 1) masking the top 10% tokens with the highest latent discrepancies $J$; 2) applying purely random masking without using the proposed $J$. The result is listed in the following table. From the table we can observe a performance drop from both ablations. Specifically, we hypothesize that the 10% masking settings hinders performance because it causes the editing process to stagnate, preventing effective guidance towards resolving large geometric differences.
>
>     |Masking Strategy  |IoU ($\uparrow$)  |Mean Chamfer Distance ($\downarrow$)  |JSD ($\downarrow$)   |Invalid Rate ($\downarrow$) |Edit Distance ($\downarrow$)   |
>     |--|--|--|--|--|--|
>     |Purely random masking 50% of primitives without P2S guidance |0.497  |0.027   |0.633   |14.7%  |22.36   |
>     |Masking top 10% w.r.t. latent discrepancy  |0.643  |0.014   |0.635   |4.8%  |19.47   |
>     |Masking above avg latent discrepancy (Our Strategy) |0.687  |0.009   |0.621   |3.1%  |16.87   |
>
>     c.	For the number of iterations. We set the maximum number of iterations as 10 throughout the current experiments. We also conduct an ablation study where the maximum number of iterations ranges between 1, 2, 5 and 10. From the table we can observe that the performance growth saturates when the iterations becomes large, which is in line with the qualitative results in Part G and Figure 4 in the supplementary material where simple modifications can be achieved at the first several iterations.
>
>     |Num. of Iterations  |IoU($\uparrow$)  |Mean Chamfer Distance($\downarrow$)  |JSD($\downarrow$)   |Invalid Rate($\downarrow$) |Edit Distance($\downarrow$)   |
>     |--|--|--|--|--|--|
>     |1  |0.512  |0.024   |0.629   |10.8%  |19.48   |
>     |2  |0.634  |0.015   |0.627   |6.8%  |17.29   |
>     |5  |0.665  |0.011   |0.626   |4.7%  |17.31   |
>     |10 (Our Original Setting) |0.687  |0.009   |0.621   |3.1%  |16.87   |
>
> Below is the discussion about the reviewer’s questions.
>
> 1.	About the timing of CADMorph. We have included a running time analysis in part E and Table 1 in the supplemental material. Besides, we provide qualitative examples about iterations needed for convergence in Part G and Figure 4 in the supplementary material, where simple edit converges at the early iterations (top example) while complex edit take more iterations to converge (bottom example).
>
> 2.	Honestly, we do not test CADMorph on complex industrial CAD models, because we currently do not have access to such data containing hundreds of CSG operations. Specifically, while DeepCAD includes up to 60 operations and Fusion360 and WHUCAD have at most 150 operations, the quantity of such complex CAD models is too scarce to form editing triplets). We think the main bottleneck lies in the lack of large-scale, high-quality datasets featuring complex CAD structures. Without exposure to such data, the MPP and P2S modules are unable to generate CAD models involving hundreds of operations. In part B of the supplementary, we discussed a potential alternatives: synthesizing complex CAD models for training by using the Boolean operations of SDFs to combine different sketch-extrusion pairs.
>
> 3.	In part B of the supplementary material, we explain the reason for choosing SDF as the first exploration and possibilities for other 3D formats. Below is a brief summary. For meshes, the graph-based representation is nuanced and the correspondence between the primitives and vertex / edge does not directly correspond to spatial occupation. For implicit representation like NeRF / Gaussian, besides the deficits mentioned in meshes, the generation time for each 3D asset is also longer than voxelized SDFs. Currently, the solution for using other 3D representations is to transform them into SDFs by leveraging other 3D models and marching cubes. We leave the innate and native integration of meshes / NeRF / Gaussians for future work.
>
> 4.	For joint training, we did not experiment with its effect due to the scarcity of editing triplets for training data, which is the motivation of CADMorph. We explored other modality interaction method. While it is not closely related to combing P2S and MPP models, we would like to share it for your reference. Specifically, we tried using natural language as the medium. The difference between the source shape (rendered from the source construction sequence) and the target shape is first summarized into natural language, and the modification is then performed using the natural language as the guidance. We made a quick PoC via the API of GPT-4o. From the experiment we can observe little difference between this mechanism and naïve GPT-4o, thus calling for the need of domain-specific foundation models and  the designed interaction strategies between the foundation models so as to trigger the abilities that they were not trained for.
> |Method  |IoU($\uparrow$)  |Mean Chamfer Distance($\downarrow$)  |JSD($\downarrow$)   |Invalid Rate($\downarrow$) |Edit Distance($\downarrow$)   |
> |--|--|--|--|--|--|
> |GPT-4o  |0.247  |0.107   |0.737   |25.1%  |21.12   |
> |Using Language Description as Intermediary  |0.177  |0.083   |0.726   |29.0%  |19.50   |
> |Ours |0.687  |0.009   |0.621   |3.1%  |16.87   |
>
> Once again, thank you for your encouraging comments, insightful feedback, as well as thought-provoking discussion.

---

> > ### Author Response · Authors · 2025-08-04
> > **Thank you for your feedback!**
> >
> > Dear reviewer,
> >
> > We sincerely thank you for your strong recognition to our work. We really enjoy discussing with you about the dataset limitation and extension to complex CAD objects, the choice of the 3D object representation, possibility of other modality interaction methods, time cost, as well as other ablation studies. Your review is really insightful and the discussion with you is indeed thought-provoking!
> >
> > Sincerely,
> >
> > Authors of paper 15168

---

> > > ### Comment · Reviewer_t3cp · 2025-08-05
> > > **Final Justifaction Made**
> > >
> > > The authors' rebuttal has addressed my previous concerns, and I therefore maintain my positive rating for this paper. I trust that the authors will revise the relevant parts as discussed. Accordingly, I recommend accepting this paper.

---

### Note · Authors · 2025-08-13

Dear Reviewers and ACs,

We sincerely thank you for all your efforts. We are delighted that reviewer t7b3 raised the score in support of our work and disagreement with qRBB, and that reviewer t3cp and ZSgx keep their positive ratings.

Strengths recognized by reviewers:

1. Clear problem setting and early mover advantage on geometry-driven parametric CAD editing (ZSgx), an important and well-motivated task (t3cp, t7b3).
2. An elegant orchestration of domain-specific foundation models (t3cp, ZSgx, t7b3), e.g. P2S cross-attention for localizing editable segments (qRBB).
3. No need for scarce editing triplets. (ZSgx, t7b3)
4. Empirical strengths over strong baselines. (t3cp, ZSgx)

Concerns addressed during rebuttal:

1. Input modality choice (t3cp). We referred to supplementary B and explained tSDF adoption. For qRBB’s suggestion on using image or sketch, we explained that tSDF is only an interface and proposed using visual foundation models to integrate them into our framework, showcasing CADMorph’s flexibility.
2. Hard cases (t3cp, Zsgx, qRBB) and BRepGen data (t7b3). We included more challenging cases in rebuttal. We noted datasets inexistence with complex CAD models and incompatibility of integrating graph-like BReps into CADMorph.
3. More ablation. We tested on iteration numbers (t3cp, ZSgx), candidate numbers (t3cp, ZSgx), masking strategies (t3cp, ZSgx), modality interaction methods and runtime analysis (t3cp, t7b3, qRBB). All concerns were resolved without further questions.

Summarized responses to reviewer qRBB:

qRBB joined the discussion late and some concerns may stem from unfamiliarity with general test-time scaling methods. We summarize our responses to aid your final decisions.

1. Novelty. Our work orchestrates the interplay between, and leverages the model intrinsics of, existing modules (MPP and P2S) to enable new test-time capabilities beyond their original purposes. This novelty is acknowledged by other reviewers (particularly t7b3). We also cite CoT and DreamFusion as precedents to show that combining existing modules at test time can be highly impactful.
2. Inference time. We report minute-level inference time across varying iterations. This is user-acceptable in LLM and test-time scaling era. We also list several engineering techniques that could cut latency by up to 40% without algorithm change.

In all, we thank reviewers and ACs for their invaluable feedback, which improves CADMorph’s quality and clarity.

Sincerely,

Authors

---

### Decision · Program_Chairs · 2025-09-17

**Decision:**

Accept (poster)

**Comment:**

This paper presents a plan–generate–verify iterative framework that cleverly integrates a P2S latent diffusion model with an LLaMA-3-based MPP, and uses cross-attention to automatically localize editable segments. This work offers a highly innovative solution to the largely unexplored task of geometry-driven parametric CAD editing, and the paper is well structured and highly readable. Although the experiments currently cover only simple parts in the CAD-Editor 2k dataset, inference still takes minutes, and deeper ablation studies or complex assembly examples are lacking, the reviewers unanimously agree that these shortcomings do not diminish the method’s overall contribution; they can be addressed in the camera-ready version by adding further experiments and efficiency analyses.

Recommendation
Accept.  This is a well-motivated work and a clear description framework. The proposed plan–generate–verify iterative method is better than LLM prompting and the baseline methods (CAD-diffuser and FlexCAD).